Young, American Indian or Alaskan Native, and born in the USA: at excess risk of starting extra-medical prescription pain reliever use?

Parker Maria A. maria.parker@uvm.edu 1 2
Lopez-Quintero Catalina 1 3
Anthony James C. 1
1 Department of Epidemiology & Biostatistics, Michigan State University , East Lansing , MI , United States of America
2 Department of Psychiatry, University of Vermont , Burlington , VT , United States of America
3 Department of Epidemiology, University of Florida , Gainesville , FL , United States of America
Patton Bob
Electronic publication date: 2018 Oct 8
Publication date: 2018
Volume: 6
Electronic Location ID: e5713
Received 2018 Jun 5; Accepted 2018 Sep 10
Copyright: ©2018 Parker et al.
Copyright year: 2018
Copyright holder: Parker et al.
License: This is an open access article distributed under the terms of the Creative Commons Attribution License, which permits unrestricted use, distribution, reproduction and adaptation in any medium and for any purpose provided that it is properly attributed. For attribution, the original author(s), title, publication source (PeerJ) and either DOI or URL of the article must be cited.
License URL: https://creativecommons.org/licenses/by/4.0/

Keywords: Opioids, Adolescents, Foreign-born, US-born, Prescription pain relievers

Funding: National Institute on Drug Abuse T32DA021129 National Institute on Drug Abuse Senior Scientist and Mentorship Award K05DA015799 Michigan State University National Institute on Drug Abuse Training Grant K01DA046715 This study was supported with funds from the National Institute on Drug Abuse (grant number T32DA021129, Parker and Lopez-Quintero); the National Institute on Drug Abuse Senior Scientist and Mentorship Award (grant number K05DA015799, Anthony); Michigan State University; and the National Institute on Drug Abuse (grant number K01DA046715, Lopez-Quintero). The funders had no role in study design, data collection and analysis, decision to publish, or preparation of the manuscript.

==============================
Background

Prescription pain reliever (PPR) overdoses differentially affect ‘American Indian/Alaskan Natives’ in the United States (US). Here, studying onset of extra-medical PPR use in 12-24-year-olds, we examine subgroup variations in rates of starting to use prescription pain relievers extra-medically (i.e., to get ‘high’ or for other reasons outside boundaries of prescriber’s intent). Risk differences (RD) are estimated for US-born versus non-US-born young people, stratified by American Indian/Alaskan Natives versus other ethnic self-identities.

Methods

Between 2002–2009, nationally representative cross-sectional samples of 12–24-year-old non-institutionalized civilians completed interviews for the US National Surveys of Drug Use and Health. Analysis-weighted annual incidence estimates, RD, and confidence intervals (CI) are from the Restricted-use Data Analysis System, an online software tool for US National Surveys of Drug Use and Health.

Results

Each year, an estimated 2.5% of 12-24-year-olds in the US start using PPR extra-medically (95% CI [2.1%–3.0%]). Estimates for the US-born (3.8%; 95% CI [3.7%–3.9%]) are larger (non-US-born: 1.8%; 95% CI [1.5%–2.0%]; RD = 2.0; p < 0.05). US-born American Indian/Alaskan Natives youths have the largest incidence rate (4.8%). Robust RD for US-born can be seen for ‘non-Hispanic White’ subgroups, and for others (e.g., ‘Cuban’, ‘Dominican’).

Discussion

Each year, one in 20 of US-born American Indian/Alaskan Natives starts using PPR extra-medically. Overdose prevention is important, but is no substitute for primary prevention initiatives for all young people. The observed epidemiological patterns can guide targeted prevention initiatives for the identified higher risk subgroups in complement with more universal prevention efforts intended to reduce incidence of first extra-medical PPR use, a crucial rate-limiting step on the path toward more serious drug involvement (i.e., progressing past initial use).

Introduction

Current indicators tracking the prescription pain reliever (PPR) epidemic in the United States (US) reflect several epidemiological processes, including increased clinician prescribing of opioid compounds (King et al., 2014). Concurrently, upward-shifting population incidence rates for starting to use PPR in an ‘extra-medical’ fashion have been seen—e.g., for feeling states such as to get high and otherwise outside boundaries intended by a prescribing clinician (Parker & Anthony, 2015).

Newer US estimates suggest recent stabilization of a previously documented upward-turning PPR epidemic curve (United States, 2016). Nevertheless, PPR dependence cases continue to require treatment resources. To illustrate, an estimated 6–9% of youths starting to use PPR extra-medically become newly incident opioid dependence cases within 12 months after first extra-medical use (Parker & Anthony, 2015). Fatal and non-fatal overdoses motivate concern and prompt public health initiatives to reduce extra-medical PPR use (Compton & Volkow, 2006).

Substantial rates of opioid PPR poisoning deaths among US residents of American Indian/Alaskan Native heritage (‘Native Americans’) generally have been neglected (Kochanek et al., 2016; Mack, 2017). While Native Americans have the highest drug overdose death rates, these rates have not yet abated and continue to increase. Overdose death rates sometimes can be traced to more rapid acceleration of extra-medical drug experiences once first use has occurred. However, the underlying explanation might be larger first-inception incidence rates for starting extra-medical PPR use—i.e., larger for Native Americans (Stanley et al., 2014).

Among those who start using PPR extra-medically, mean and median age values are well under 25 years, and initiation rates for 12–17-year-olds do not differ appreciably from those in young adulthood (e.g., 18-24-year-olds; (Parker & Anthony, 2015; United States, 2016). Hence, it is important to examine estimates focused on 12–24-year-olds.

Quite plausibly, US-born young people in this age range might be generally more likely to start using PPR when compared with their non-US-born immigrant peers, and rates of starting extra-medical PPR use might be greater for US-born Native Americans versus those born outside the US (United States, 2013). For this reason, we organized the analysis approach to compare US versus non-US born populations, with a focus on US-born versus non-US born American Indian/Alaskan Natives (e.g., belonging to Central or South American Indian groups). By focusing on US young people’s PPR risk experiences in 2002–2009, we leave an opportunity for evaluation of reproducibility when more recent data become available for 2010 through 2017. Our hope is that the results, once replicated in newer datasets from 2010–2017, will help guide plans for ending the US opioid epidemic (Kolodny & Frieden, 2017).

Materials & Methods

Study population, participants, & data collection

The study population consists of non-institutionalized US community residents age 12–24-years-old, as sampled, recruited, and assessed using Institutional Review Board approved protocols for National Surveys on Drug Use and Health (NSDUH). These surveys involved drawing new nationally representative samples each year, 2002–2009, generally with >70% participation levels (United States, 2013). Publications and online reports provide detailed descriptions of NSDUH methods and the no longer available Restricted-use Data Analysis System (RDAS) datafiles that enabled stratification by US birthplace (e.g.,  https://www.samhsa.gov/data/all-reports; United States, 2013; Vsevolozhskaya & Anthony, 2014).

Unlike other datasets, the formerly available RDAS datafiles for 2002–2009 covered more than 200,000 12–24-year-olds with detailed variables on ethnic self-identification (ESI; beyond the limited race-ethnicity subgroups of US federal research: https://grants.nih.gov/grants/funding/women_min/inclusion_ims.htm) and US-born status. Participation involved completion of audio computer self-interviews (ACASI) with standardized multi-item modules.

Measures

The ACASI items on extra-medical PPR use identify never-users, past-onset users, and ‘past-year initiates,’ as described in detail previously (Parker & Anthony, 2015). ACASI ESI items ask “Which of these groups describes you?” with detailed responses: Mexican/Mexican American/Chicano, Puerto Rican, Central or South American, Cuban/Cuban American, Dominican (i.e., from the Dominican Republic), Chinese, Filipino, Indian (Asian), Vietnamese, Korean, Japanese, Other Asian (non-specified), non-Hispanic White, non-Hispanic Black/African American, American Indian or Alaskan Native (herein abbreviated as ‘Native American’), Native Hawaiian, Other Pacific Islander (excluding Native Hawaiian), Other (non-specified).

Analysis

Analysis-weighted incidence rates reported here are from RDAS variables designating a numerator (‘past-year initiates’ or newly incident users with extra-medical PPR onset within 12 months before assessment), and a denominator of all ‘at risk’ of doing so. Respondents with no use of extra-medical PPR (in the lifetime) as well as the newly incident users were counted in the denominators of this study’s incidence rates as members of the ‘at-risk’ population. The denominator excluded past-onset users because they are no longer candidates for becoming newly incident users.

In this fashion, annual incidence rates were formed for 18 ESI subgroups, and risk differences (RD) were calculated to contrast US-born and non-US-born immigrant experiences. RD estimates are formed by subtracting each subgroup-specific incidence rate for non-US-born immigrants from the corresponding US-born estimate. RDAS provided analysis weights appropriate for pooled analyses of 2002–2009 data bringing age and sex into balance with US Census distributions via Taylor series linearization standard errors (United States, 2013; Vsevolozhskaya & Anthony, 2014). We derived 95% confidence intervals (CI) and evaluated null hypotheses at RD = 0 with alpha at 0.05.

Results

Table 1 depicts unweighted subgroup sample sizes, detailed annual incidence estimates for ESI and birthplace subgroups, and RD. The overall incidence rate estimate for US 12–24-year-olds is 2.5% (95% CI [2.1%–3.0%])—i.e., an estimated two to three newly incident users per 100 12–24-year-olds (unweighted denominator n = 173,961). Corresponding estimates for non-US-born immigrants versus the US-born were 1.8% (95% CI [1.5–2.0]) versus 3.8% (95% CI [3.7–3.9]), respectively, with RD = 2.0 (p < 0.05; Table 1). Evaluated for all 36 subgroups under study, the largest observed estimate is 4.5%, seen for Native American 12–24-year-olds (n = 2,031), with excess risk concentrated among the US-born (4.8%; p < 0.05; Table 1).

Table 1 Estimated annual incidence of extra-medical prescription pain reliever use for 12-24-year-olds in the United States (US), stratified by ethnic self-identification subgroups and birthplace.a

Ethnic self-identification subgroups		Total	US-born	Non-US-born	US-born minus non-US-bornb	
	Unweighted nc	%	SE	%	SE	%	SE	Risk difference	SE	
Overall (12–24 Total)	173,961	2.5	0.2	3.8	0.1	1.8	0.1	2.0	0.1	
Native American (American Indian/Alaskan Native)	2,031	4.5	0.5	4.8	0.4	1.8	0.4	3.0	0.6	
Mexican/Mexican American/Chicano	16,104	2.8	0.1	3.2	0.2	1.7	0.2	1.5	0.3	
Puerto Rican	2,126	3.0	0.4	3.1	0.4	2.5	0.9	0.6	1.0	
Central/South American	2,984	1.9	0.3	2.2	0.4	1.5	0.4	0.7	0.5	
Cuban/Cuban-American	1,263	3.4	0.5	4.1	0.6	0.9	0.5	3.2	0.8	
Dominican (Dominican Republic)	696	2.4	0.6	3.4	0.8	0.4	0.3	3.0	0.9	
Chinese	1,357	1.5	0.3	1.4	0.4	1.7	0.5	−0.3	0.7	
Filipino	736	1.8	0.5	2.1	0.5	1.4	1.0	0.7	1.1	
Indian (Asian)	1,534	1.4	0.3	1.8	0.5	1.1	0.4	0.7	0.6	
Vietnamese	694	1.3	0.4	1.6	0.6	0.9	0.6	0.7	0.8	
Korean	707	1.8	0.5	2.3	0.9	1.4	0.6	0.9	1.0	
Japanesed	527	2.5	0.7	2.9	0.9	–	–	–	–	
Other Asian (Non-Specified)	1,089	0.8	0.3	1.1	0.4	0.5	0.3	0.6	0.5	
Non-Hispanic White	109,219	4.1	0.1	4.3	0.1	1.9	0.2	2.4	0.2	
Non-Hispanic Black	26,563	2.2	0.1	2.2	0.1	1.8	0.4	0.4	0.4	
Native Hawaiiand	670	3.5	0.7	–	–	–	–	–	–	
Pacific Islander (Excluding Native Hawaiian)	782	2.8	0.6	3.0	0.7	2.3	1.2	0.7	1.4	
Other (Non-Specified)	4,879	2.2	0.2	2.3	0.3	2.0	0.4	0.3	0.4	
Notes.

a Parker & Anthony explain the history of the concept of ‘extra-medical’ use since this term was introduced (2015).

b Bolding denotes statistical significance at the alpha = 0.05 level.

c Unweighted RDAS subgroup size approximations with overall sum at risk for starting extra-medical prescription pain reliever use. RDAS output yields weighted estimates but does not disclose unweighted cell counts, which we have derived using an approximation method (Vsevolozhskaya & Anthony, 2014).

d Not all incidence rates for Japanese and Native Hawaiian by birthplace were estimable due to too few newly incident users.

Table 1 also shows several ESI subgroups with distinctively large incidence rates, including non-Hispanic Whites and the Hispanic subgroups with ESI as ‘Cuban’ or as ‘Dominican.’ Distinctively lower incidence rates are seen for ‘Chinese’ as well as ‘Other Asians’ (i.e., ‘not otherwise specified as Asian’). The estimated RD is 2.0% for all 12–24-year-olds (SE = 0.1), versus 3.0% for those of Native American heritage (SE = 0.6; Table 1, RD Estimates).

Figure 1 depicts these incidence estimates and RD in two dimensions. The x-axis arranges incidence estimates for US-born of each subgroup from smallest (Other Asian, 1.1%) to largest (Native American, 4.8%). The y-axis presents estimated RD and 95% CI, showing the just-mentioned excess risk for the US-born among 12–24-year-olds generally (RD = 2.0%). Non-null RD are seen for non-Hispanic Whites (RD = 2.4%), and corresponding ESI subgroups of Native Americans (RD = 3.0%); Cubans (RD = 3.2%), Dominicans (RD = 3.0%), and Mexicans (RD = 1.5%), for which each 95% CI fails to touch the x-axis at y = 0 (Fig. 1). The preponderance of Western Hemisphere subgroups is noteworthy in this list, as is under-representation of Asians. For other subgroups, RD estimates are smaller and statistically imprecise, but larger rates generally are seen for US-born. The Chinese young people provide an unexplained exception to the general rule, but we note that the 95% CI for this estimate crosses the x-axis at y = 0, and denotes an approximate p > 0.05.

Figure 1 Estimated differences in the annual incidence rates and risk differences for extra-medical prescription pain reliever use among 12–24-year-olds by ethnic self-identification subgroups and birthplace.

Incidence estimates and RD are shown in two dimensions. The x-axis arranges incidence estimates for US-born of each subgroup from smallest to largest. The y-axis presents estimated RD and 95% CI.

Discussion

It can be useful to conceptualize drug involvement across a spectrum of experiences that range from an initial chance to try a drug and onward toward more serious experiences such as addictive states or overdose. In this work, we focus on early stages of PPR involvement among young people, namely, the first occasion of extra-medical PPR use. These novel epidemiological estimates describe but cannot explain how this country’s US-born young people of Native American heritage have such large incidence rates, versus lower non-US-born rates, as first documented here. However, the observed larger overdose rates for this US subgroup might be traced, at least in part, to a failure of current programs to prevent extra-medical PPR use (before it starts) among young people of Native American heritage.

Paired with the high rates of clinician prescribing in the US (Manchikanti et al., 2012), it may be that opioid public health initiatives also are not reaching many of the Tribal communities. Targeted prevention initiatives for Native Americans such as school screenings or community education efforts complemented with more universal prevention effects could reduce the incidence of first extra-medical PPR use (e.g., The Truth about Opioids: opioids.thetruth.com).

These annual incidence rate estimates offer a reminder that preventive medicine must complement opioid dependence treatment and overdose services (Kolodny & Frieden, 2017). Epidemiological estimates of this type also can be useful as differentially affected population subgroups may have been overlooked.

Before detailed discussion of these results, several of the more important study limitations merit attention, including the cross-sectional NSDUH design, the self-reported nature of the data, and the somewhat historical nature of the 2002–2009 datasets. However, if we can assume that ESI and US-born status qualify as variables that are relatively time-invariant, and do not change after onset of extra-medical PPR use, then the research design does not create inferential difficulties that epidemiologists face when time-varying and drug-responsive characteristics are investigated at cross-section. We do note, however, some smaller sample sizes where the RD could not be estimated (i.e., Japanese and Native Hawaiian). As for the historical nature of these data, we had anticipated release of NSDUH data from 2010 onward, but the US federal agencies have delayed release of these more recent datasets. Datasets from surveys completed during 2010–2017 should make it possible to replicate this study’s estimates, or to disconfirm them.

This epidemiological study represents a departure from prior studies focused on prevalence of PPR use—that is, being a user. In contrast, our study concerns incidence or becoming an extra-medical PPR user for the first time. During 2002–2009, there is clear evidence that US-born subgroups generally experienced excess risk as compared with non-US-born immigrants. ESI subgroups from the Western Hemisphere serve best to illustrate this variation in risk experience. In ESI subgroups from Asian countries, smaller incidence rates and a smaller US-born excess generally is seen.

We are hopeful that these initial clues about uneven distribution of PPR incidence rates will set up new lines of study, including investigation of male–female differences within implicated subgroups as well as potential effects of time since immigration to the US. Future research should explore whether the effects of time since immigration might be modulated by age at immigration or by within-household characteristics such as language spoken at home.

We have left the opportunity for evaluation of reproducibility with newer NSDUH data from 2010–2017, in extension of what we have provided as estimates from mid-epidemic 2002–2009 surveillance analyses. For this reason, we decided to seek publication of our initial 2002–2009 estimates, and we are hopeful about confirmation with data from more recent years. Nonetheless, we believe it will take a combined qualitative and mixed methods research approach to illuminate generating mechanisms for the observed excess PPR rates, which might include historical or social trauma, as well as norms of US-born Native American youth (Stanley et al., 2014; Stanley, Swaim & Dieterich, 2017).

Much remains to be learned on the topic of drug involvement in relation to immigrant health. Here, we provide preliminary evidence of what is likely to become recognized as a more important source of variation in the health of subpopulations defined by immigrant status—namely, the degree to which immigrants retain the drug-taking practices of their countries of origin versus adoption of the drug-taking norms in their new neighborhoods.

Conclusion

Looking into the earliest stages of drug involvement, here with emphasis on first extra-medical drug use, we see that young people born in the US are at greater risk of starting to use PPR extra-medically, as compared with non-US-born immigrant peers. Native Americans, an important but often neglected subgroup of the US population, apparently have the largest annual incidence rate for onset of extra-medical use of PPRs. With focus upon first-time drug users, this work’s epidemiological estimates are especially pertinent for public health workers and leaders who are responsible for primary prevention of use.

The authors wish to thank Holly Vredevelt for her help with the tables in the early stages of this manuscript.

Additional Information and Declarations

Competing Interests

Author Contributions

Data Availability

The authors declare there are no competing interests.

Maria A. Parker performed the experiments, analyzed the data, contributed reagents/materials/analysis tools, prepared figures and/or tables, authored or reviewed drafts of the paper, approved the final draft.

Catalina Lopez-Quintero conceived and designed the experiments, performed the experiments, contributed reagents/materials/analysis tools, prepared figures and/or tables, authored or reviewed drafts of the paper, approved the final draft.

James C. Anthony conceived and designed the experiments, contributed reagents/materials/analysis tools, authored or reviewed drafts of the paper, approved the final draft.

The following information was supplied regarding data availability:

This study was conducted with data from the National Surveys on Drug Use and Health data from 2002–2009, which were available to the public via an Inter-university Consortium for Political and Social Research website (i.e., https://www.icpsr.umich.edu/icpsrweb/NAHDAP/series/64).

ICPSR hosted the NSDUH online data analysis system known as the Restricted-Use Data Analysis System (RDAS). RDAS was used to produce our study estimates. RDAS made NSDUH data available via an online analysis interface designed to enable analysis-weighted contingency tables, but protects research participants by disabling data download features. For this reason, others can replicate our RDAS analyses, but no one (outside the government) has access to the confidential raw data.

While this paper was under review, the federal government re-enabled the RDAS in a series of steps that began with release of the P-DAS (Public-Use Data Analysis System), which has limited variables available: https://pdas.samhsa.gov/, and the Spring 2018 release of a new RDAS, which has more limited functionality than the original RDAS. We are exploring whether the new iteration of RDAS will make it possible to re-examine the relationships under study in this work. At present, it appears the new version of RDAS does not yet have the capacity for replication of the estimates reported in this paper.

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
