# Peer review of "Young, American Indian or Alaskan Native, and born in the USA: at excess risk of starting extra-medical prescription pain reliever use?"

_PeerJ, doi:10.7717/peerj.5713_

## Round 0.1 · original submission · Minor Revisions

Thank you for this submission which has been well received by our reviewers. Some minor revisions are required before we can consider your paper for publication.

Reviewer 1 ·

Basic reporting

Overall, the manuscript reads well, but there are typos, missing words and use of idioms, which take away from the quality of the manuscript, e.g. line 129 ‘Figure 1 shows depicts these…’, line 179 ‘scratch the surface’ and line 189 missing word (selected examples).

Experimental design

See below.

Validity of the findings

No comments.

Additional comments

Thank you for the opportunity to review this interesting manuscript, offering further data and insights into the dire situation in the US of high rates of opioid overdose deaths. I think the focus on sub-populations is very important and perhaps somewhat overlooked in previous work.

Whilst the authors briefly discuss the age of the data (some data waves being more than 15 years old) in lines 170 to 174, reading the Introduction, Results and Discussion, might easily leave the impression that this is new data, e.g. line 147: ‘these new annual incidence rates estimates’ and line 143 to 144: ‘this country’s US-born young people…have such large incidence rates…’ (They had in this dataset but do they today?). I would hope that the authors would make this clearer in the Introduction and Discussion. What’s the value of the dataset? It is historic insight, does it apply to the current situation, and if so, how?

Minor comments:

Lines 39 to 42: Please elaborate as to what you have in mind in regard to ‘targeted prevention initiatives’ and ‘more universal prevention efforts’.

Line 42: The meaning of ‘more serious drug involvement’ is not clear. As the authors convincingly argue, opioid overdose deaths are very serious, and hence this sentence comes across as a self-contradiction. Perhaps the authors are referring to PO misuse transitioning into heroin use or use of fentanyl contaminated fake medicines?

Line 44: Epidemic of what?

Line 47: The meaning of the term ‘hazard-laden’ is not clear. Is it needed considering that the authors provide a good and useable definition of ‘extra-medical’ PPR use? Furthermore, there would be examples of ‘extra-medical’ use that would not be ‘hazard-laden’.

Line 51: IBID. Epidemic curve of what?

Line 54: Are casualties not the same as fatal overdoses?

Line 59: What is meant by ‘have been neglected’? Gap in research and/or treatment?

Line 81: Please provide a brief explanation for the use of this dataset (see general comments above).

Line 89: Which questions were used to obtain data on PPR use?

Line 115: What is the gender distribution in the sample? Distribution according to US regions? Perhaps it would be useful to include a table with descriptive analysis to characterise the sample?

Line 142: What is meant by ‘PPR involvement’?

Line 178: What is meant by ‘drug involvement’?

Reviewer 2 ·

Basic reporting

This is an outstanding paper, clearly written, with a enough information and references to provide background support to the authors' decisions.

Experimental design

This is novel research that, in my opinion, makes the best use of the NSDUH data. Rarely investigators study incidence of drug involvement. This has many advantages because many threats to validity are avoided. The use of a probability sample with appropriate statistical methods are also part of the strengths of this study.

Validity of the findings

The study provides novel and strong new insights to support new research, and the authors offer ideas about potential areas for new studies. The only factor that limits excitement is that the analyses are limited to the period of 2002-2009, and it would be useful to have more recent data. It is not clear why the authors chose this particular period, and the assumptions behind the idea that a replication with a 2010-2017 period should be informative. Clearly, the overdose crisis of PPR has received much more attention in the past years, hopefully modifying the conditions that governed during the 2002-2009 period.

Additional comments

This is an excellent report. Publication of this article may be useful to support increased resources for programs serving the populations at higher risk identified here.

---

## Round 0.2 · accepted · Accept

Thank you for the revised version. The reviewers felt that you had addressed all of their concerns, however they have suggested that you proof read the final version to check the phrasing of some of your revised content. This will not require a re-submission and can be done while in Production.

# Reviewer 2 ·

Basic reporting

The authors have successfully edited the document to provide more clarity.

Experimental design

The design has not been changed and was already acceptable.

Validity of the findings

Idem. The authors have done a good job in revising and amending the manuscript for clarity, which allows to realize the strengths and limitations of the research.

Additional comments

While most of the issues have been resolved, the manuscript can be improved by one last editorial revision. Some of the phrases that were added in the newest version read awkward (e.g., page 8, line 190).